# Mitigation of salinity stress in yarrow (*Achillea millefolium* L.) plants through spermidine application

**Sajedeh Alijani**[1], **Mohammad-Reza Raji**[1]*, **Zohreh Emami Bistgani**[2], **Abdollah Ehtesham Nia**[1], **Mostafa Farajpour**[3]*

**1** Department of Horticulture, College of Agriculture, Lorestan University, Khorramabad, Iran, **2** Isfahan Agricultural and Natural Resources Research and Education Center, Agricultural Research Education and Extension Organization (AREEO), Isfahan, Iran, **3** Crop and Horticultural Science Research Department, Mazandaran Agricultural and Natural Resources Research and Education Center, AREEO, Sari, Iran

\* m.farajpour@areeo.ac.ir (MF); raji.m@lu.ac.ir (MRR)

**Data Availability Statement:** All relevant data are within the manuscript and its Supporting Information files.

## Abstract

This study investigated the mitigating effects of spermidine on salinity-stressed yarrow plants (*Achillea millefolium* L.), an economically important medicinal crop. Plants were treated with four salinity levels (0, 30, 60, 90 mM NaCl) and three spermidine concentrations (0, 1.5, 3 µM). Salinity induced electrolyte leakage in a dose-dependent manner, increasing from 22% at 30 mM to 56% at 90 mM NaCl without spermidine. However, 1.5 µM spermidine significantly reduced leakage across salinities by 1.35–11.2% relative to untreated stressed plants. Photosynthetic pigments (chlorophyll a, b, carotenoids) also exhibited salinity- and spermidine-modulated responses. While salinity decreased chlorophyll a, both spermidine concentrations increased chlorophyll b and carotenoids under most saline conditions. Salinity and spermidine synergistically elevated osmoprotectants proline and total carbohydrates, with 3 µM spermidine augmenting proline and carbohydrates up to 14.4% and 13.1% at 90 mM NaCl, respectively. Antioxidant enzymes CAT, POD and APX displayed complex regulation influenced by treatment factors. Moreover, salinity stress and spermidine also influenced the expression of linalool and pinene synthetase genes, with the highest expression levels observed under 90 mM salt stress and the application of 3 µM spermidine. The findings provide valuable insights into the responses of yarrow plants to salinity stress and highlight the potential of spermidine in mitigating the adverse effects of salinity stress.

## Introduction

Yarrow (*Achillea millefolium* L.) is a perennial member of the Compositeae family. It typically reaches a height of 20 to 90 centimeters and features delicate, feathery leaves, hairy long stems, and clusters of white or yellowish-white flowers at the apex of the stem [1]. All parts of the plant possess a distinctive aroma and a bitter taste [2].

Yarrow is employed for various purposes, including wound healing, alleviating digestive issues, reducing respiratory infections, improving skin conditions, and treating liver diseases

**Funding:** The author(s) received no specific funding for this work.

**Competing interests:** The authors have declared that no competing interests exist.

[3–5]. It exhibits mild sedative, anti-inflammatory, anti-rheumatic, antispasmodic, antiseptic, soothing, astringent, and digestive properties [5, 6].

Soil or water salinity poses a significant stress factor in arid and semi-arid regions, severely impeding plant growth and performance. Typically, plants respond to salinity by stunting their growth. Leaves accumulate salt, leading to premature aging, reduced production of assimilates for growth regions, and consequently, diminished plant growth [7]. Salinity stress primarily diminishes photosynthesis in plants due to a decrease in water potential. Salinity stress rapidly affects photosynthesis and cell growth, both of which are among the earliest processes affected [8]. Salinity stress induces an overall reduction in growth and plant performance, disrupting essential components of photosynthesis such as photosystem II and the enzyme ribulose-1,5-bisphosphate carboxylase/oxygenase [9].

Salinity stress, similar to other environmental stresses, triggers the accumulation of reactive oxygen species (ROS), including superoxide, hydrogen peroxide, and hydroxyl radicals, within plant cells. These ROS can cause damage to membrane lipids, proteins, and nucleic acids. The effect of free radicals on lipid peroxidation, protein coagulation, DNA mutations, and oxidation of other biological molecules has been extensively demonstrated [10]. Researchers commonly employ membrane stability index and lipid peroxidation as indicators to assess salinity damage and tolerance in plants, with salinity resulting in reduced membrane stability [11].

In higher plants, the enzymatic antioxidant system plays a crucial role in combating ROS. This system includes enzymes such as catalase, superoxide dismutase, polyphenol oxidase, and ascorbate peroxidase, as well as low molecular weight scavengers such as ascorbate, glutathione, and proline. These components effectively eliminate ROS generated under stress conditions and constitute the primary line of defense against ROS in various cellular compartments [12, 13]. Catalase and ascorbate peroxidase are particularly important antioxidants as they break down hydrogen peroxide into water and oxygen [14]. Ascorbate peroxidases play a critical role in eliminating ROS and safeguarding cells from their detrimental effects in algae and higher plants [15].

Furthermore, environmental stresses can disrupt osmotic regulators in different plant organs [16]. Proline, a significant osmolyte, plays a vital role in regulating cell osmotic pressure under environmental stress. Increased proline production counteracts the effects of salinity disturbances on cellular processes [17]. In addition to its osmotic regulation function, proline acts as a protective agent against enzymes and scavenges free radicals, offering defense against oxidative stress [18]. Moreover, proline helps maintain cytoplasmic water-holding capacity, thereby preserving macromolecules such as proteins and specific cytoplasmic and mitochondrial enzymes, preventing their undesirable aggregation or fragmentation [19].

In the realm of plants, the unparalleled protective effects of monoterpenes and other volatile isoprenoids have been stimulated by adverse conditions such as oxidative stress and high temperature, as documented by Loreto and Schnitzler [20]. The remarkable release of predominant monoterpenes, such as linalool and limonene, in response to heat-induced stress indicates a potent capacity to counteract oxygen-related damage. It has been suggested that sesquiterpenes can fulfill specific functions in plant cells to alleviate the detrimental effects of stress through antioxidant-based mechanisms, highlighting their effectiveness in scavenging reactive oxygen species (ROS) [21]. Moreover, recent studies suggest that excessive production of monoterpenes and sesquiterpenes may possess antioxidant properties similar to protective agents against $H_2O_2$. They are regarded as a crucial component of non-enzymatic oxidative defense systems during water stress, providing safeguarding against oxidative stress in basil [22]. The importance of monoterpenes for salinity and drought stress lies in their ability to act as powerful antioxidants, neutralizing harmful reactive oxygen species and protecting plant cells from oxidative damage. Monoterpenes, along with other volatile isoprenoids, are induced

in response to abiotic stresses such as salinity and drought, providing a defense mechanism for plants. The release of major monoterpenes like linalool during heat-induced stress suggests their role in mitigating the negative effects of high temperatures. Furthermore, sesquiterpenes have been found to contribute to stress response in plants by scavenging ROS. The overproduction of monoterpenes and sesquiterpenes has been associated with antioxidant properties, similar to those of protective agents against hydrogen peroxide. This indicates their potential as key components in non-enzymatic oxidative defense systems, particularly during water stress, offering protection against oxidative stress in plants like basil.

Polyamines, including spermine, spermidine, and putrescine, are low-molecular-weight organic polycations that can interact with nucleic acids, proteins, membrane phospholipids, and cell wall components under physiological acidity conditions. These compounds serve as initiators of signaling cascades or growth regulators, contributing to resistance against adverse factors [23]. Polyamines play diverse roles in various physiological processes, including photosynthetic activity, membrane stabilization, enzyme activation, stimulation of cell division, cell growth and development, DNA and protein synthesis, control of root branching, embryogenesis, flower development, organ growth, seed germination, and responses to both biotic and abiotic stresses [24, 25]. The polycationic nature of polyamines under physiological pH is a critical characteristic that influences their biological activities [26]. Polyamines can act as stress messengers in different environmental conditions, competing with ethylene, a common precursor called S-adenosylmethionine. Due to their ability to neutralize reactive oxygen species generated under biological and non-biological stress conditions, polyamines are recognized as stress-protective compounds [27].

The aim of this study was to investigate the effects of spermidine, a polyamine, and salinity stress on yarrow plants. Specifically, the study aimed to assess the physiological and molecular responses of yarrow plants to different levels of salinity stress and spermidine concentration.

## Material and methods

A factorial experiment based on a completely randomized design with four repetitions, was conducted in the greenhouse of the Agricultural Research and Education Center in Isfahan Province. The seedling of the yarrow plants sourced from the medicinal plants field of the same research center. The seeds were transplanted to the greenhouse in early December, subject to daily temperatures ranging between 10 and 15 degrees Celsius, coupled with regular irrigation. The experimental factors included salinity, varying sodium chloride concentrations at 30, 60, and 90 mM, and the application of spermidine via foliar spraying at concentrations of 1.5 and 3 μM. In each irrigation event, a total of 200 cc of saline solution was utilized for watering the pot. Salinity stress was incrementally introduced over a two-week period. The foliar application of spermidine commenced prior to salinity stress, initiating two weeks after the plants were established in the greenhouse, occurring once every two weeks throughout the stress period.

### Electrolyte leakage

Three leaf discs with a diameter of 5.0 cm were prepared from each plant. The discs were incubated in distilled water overnight at 25°C, after which the initial electrical conductivity (EC1) was measured. The samples were then autoclaved at 120°C for 30 minutes to release total electrolytes from disrupted cells. The final electrical conductivity (EC2) was measured after cooling to 25°C. The electrolyte leakage was calculated using following equation [28].

ELR (%) = EC1/EC2 × 100

## Photosynthetic pigments

Photosynthetic pigments were measured using a spectrophotometer, and the light absorption at three wavelengths, 470, 665, and 662 nanometers, was recorded. Finally, the pigment levels were calculated based on the following equation. In this equation, A661.6 (absorbance at 662 nanometers), A644.8 (absorbance at 645 nanometers), and A470 (absorbance at 470 nanometers) were taken into account.

Chl a = $11.24 \times (A_{661.6}) - 2.04 \times (A_{644.8})$

Chl b = $20.13 \times (A_{644.8}) - 4.19 \times (A_{661.6})$

Chl a+Ch b = $7.05 \times (A_{661.6}) + 18.09 \times (A_{644.8})$

Cartenoide = $(1000 \times A_{470} - 1.9 \times Ch\ a - 63.14 \times Ch\ b)/214$

## Proline content

Fresh leaf tissue (0.2 g) was homogenized in 10 mL of 3% sulfosalicylic acid. The homogenate was centrifuged at 13,000 rpm for 10 minutes at 4°C. Then, 2 mL of the supernatant was reacted with 2 mL ninhydrin reagent and 2 mL glacial acetic acid by heating at 100°C for 1 hour in a water bath. The reaction mixture was cooled and 4 mL toluene was added to extract the proline-ninhydrin complex. Absorbance of the toluene fraction was measured at 520 nm using a spectrophotometer. Proline concentration was determined from a standard curve and expressed as milligrams proline per gram fresh weight [29].

## Soluble carbohydrates

To extract soluble carbohydrates, 0.2 grams of leaf tissue were heated with 10 milliliters of 95% ethanol in a water bath at 80°C for one hour. Following this, 1 milliliter of 5% phenol and 5 milliliters of concentrated 98% sulfuric acid were added to the extract. The absorbance at a wavelength of 483 nanometers was then measured using a spectrophotometer. The amount of extracted carbohydrates was determined in micrograms of glucose per gram of fresh weight using a standard glucose curve.

## POD activity

Peroxidase activity was measured based on the enzymatic formation of tetraguaiacol from guaiacol and hydrogen peroxide. Leaf tissue was homogenized in buffer and centrifuged to obtain a supernatant containing the enzyme extract. The reaction mixture consisted of the enzyme extract, guaiacol, hydrogen peroxide and buffer. Absorbance at 470nm was recorded over 3 minutes at 15 second intervals against a blank. The amount of tetraguaiacol formed was quantified using its molar extinction coefficient of 26.6 $mM^{-1}cm^{-1}$. Peroxidase activity was expressed as µmoles of tetraguaiacol formed per minute per mg of protein. Protein concentration was determined by the Bradford assay. The specific activity was calculated using absorbance readings, extinction coefficient and slopes of the linear portions of the absorbance curves [30].

## CAT activity

The catalase enzyme activity was measured at a temperature of 25 degrees Celsius using a spectrophotometer at a wavelength of 240 nanometers. The solutions and materials used included 3000 microliters of 50 mM phosphate buffer (pH 7), 5 microliters of 30% $H_2O_2$, and 50 microliters of enzyme extract. The enzyme activity was recorded for 5 minutes at 20-second intervals. Finally, the specific enzyme activity was reported as micromoles of $H_2O_2$ decomposed per minute per milligram of protein [31].

## APX activity

About 0.1 gram of leaf sample was homogenized with 1000 microliters of sodium phosphate buffer containing EDTA and PVP. The resulting solution was centrifuged at 14,000 rpm for 20 minutes at a temperature of 4 degrees Celsius. 1500 microliters of acid-free extraction buffer, 300 microliters of buffer containing ascorbic acid, and 3 microliters of oxygenated water were mixed with 50 microliters of the supernatant and read at a wavelength of 290 nanometers using a spectrophotometer. The enzyme activity was calculated as micromoles per minute per gram fresh weight using following equation [30].

## RNA extraction and cDNA synthesis

To extract total RNA, 200 milligrams of leaf samples from treated and control plants of the species were collected in three biological replicates and immediately placed in liquid nitrogen. The leaves were then powdered using a mortar and extracted using the Takara RNA extraction kit protocol. To remove genomic DNA contamination, the samples were treated with DNase I (Free-RNase R DNase). The quantity and quality of the extracted RNA were evaluated using a NanoDrop instrument and 1% agarose gel electrophoresis. The cDNA strands were synthesized using the Takara cDNA synthesis kit, reverse transcriptase enzyme, and Oligo-dT primers according to the manufacturer's instructions.

## Real-time PCR reactions

Primers for the pinene synthase and linalool synthase genes were synthesized according to the study conducted by Poorraskari et al. [32] (Table 1). In brief, the reaction mixture contained 3 microliters of cDNA, 1 microliter (50 nanograms) of each forward and reverse primer, 10 microliters of master mix containing SYBR Green (Takara, Japan), and 5 microliters of sterile distilled water. The PCR reaction conditions were as follows: an initial denaturation at 95˚C for 2 min, followed by 45 cycles of 95˚C for 15 Sec, 55˚C for 20 Sec, and 72˚C for 20 Sec. The real-time PCR reaction was performed in a volume of 20 microliters, in technical duplicates, and for each biological replicate. The data were analyzed using the $2^{-\Delta\Delta CT}$ threshold cycle method. The threshold cycle values for different samples were normalized using a reference gene (Actin), and the relative difference in gene expression levels was calculated using the LightCycler® 96 SW 1.1 software (Roche, Life Science).

## Statistical analysis

The obtained data were analyzed using the statistical software SAS. The means of the data were compared using the least significant difference (LSD) test at a 5% significance level, and the graphs were plotted using Excel software.

**Table 1. Sequence of forwards and reverse primers used in qRT-PCR reaction.**

| Gene | Primer | Primer sequence (5' - 3') | Annealing temp. | Length (bp) |
|---|---|---|---|---|
| Actin | F | TGGAATGGAAGCTGCTGGTA | 58 | 183 |
| | R | TTGATCTTCATGCTGCTCGG | 58 | |
| Pinene | F | CAAGGTGGTGGAAAGAAAC | 55.5 | 181 |
| | R | GTACCATACACATCATAAACATC | 54.5 | |
| Linalool | F | ATATCAGGACCCGTAGCACTTATG | 60.9 | 139 |
| | R | CCAAGTCATCAGTAAGTCGGAAG | 59.7 | |

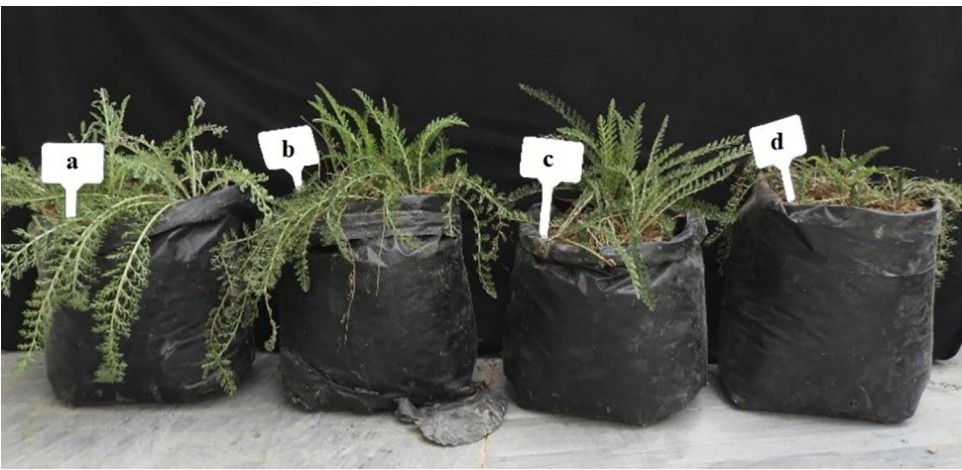

**Fig 1. Effect of spermidine and salinity on plant growth in *Achillea millefolium*.** (a) Control (0 mM NaCl, 0 μM spermidine); (b) 3 μM spermidine with 0 mM NaCl; (c) 3 μM spermidine with 90 mM NaCl; and (d) 90 mM NaCl with 0 μM spermidine.

## Results

### Spermidine reduces electrolyte leakage under salinity stress

The ANOVA results demonstrated a significant impact of salinity stress, spermidine, and their interaction at a 1% probability level on plant electrolyte leakage.

As shown in Fig 1, plants exposed only to salinity stress (d) displayed severe wilting compared to control (a), while spermidine alleviated most of the deleterious effects of salt when combined with NaCl treatment (c), indicating its protective function against osmotic stress. The findings also unveiled a notable increase in electrolyte leakage corresponding to elevated salinity stress (Fig 2). Electrolyte leakage in control plants ranged from 42.11–49.75% depending on spermidine concentration (Fig 2). Under non-stressed conditions, 1.5 and 3 μM spermidine resulted in moderately higher leakage of 46.24 and 49.75% respectively compared to the control. Salinity stress steadily increased electrolyte leakage in a dose-dependent manner, with the highest value of 56% observed at 90 mM without spermidine treatment. Application of 1.5 μM spermidine significantly reduced electrolyte leakage across all salinity levels compared to untreated stressed plants. At 30, 60 and 90 mM NaCl, 1.5 μM spermidine lowered leakage by 1.35, 7.99 and 11.2% respectively. On the other hand, 3 μM spermidine was less effective in mitigating the rise in leakage caused by salinity. These findings indicate that 1.5 μM spermidine is more efficient in maintaining plasma membrane integrity under salinity stress. Overall, the results demonstrate that spermidine enhances membrane stability and decreases electrolyte discharge from cells under salinity stress conditions, especially at the 1.5 μM concentration. This suggests spermidine protects cellular functions by reducing oxidative damage to membranes.

### Spermidine increase chlorophyll b and carotenoid, while reduce chlorophyll a under salinity stress

The results of the variance analysis indicated a significant effect of salinity stress, spermidine, and their interaction on chlorophyll a, b, and carotenoid contents (P < 0.01). According to the results, salinity stress led to a reduction in the levels of chlorophyll a, b, and carotenoid (Fig 3). As shown in Fig 3, the highest levels of chlorophyll a (28.49 mg/g FW), b (16.23 mg/g FW) and

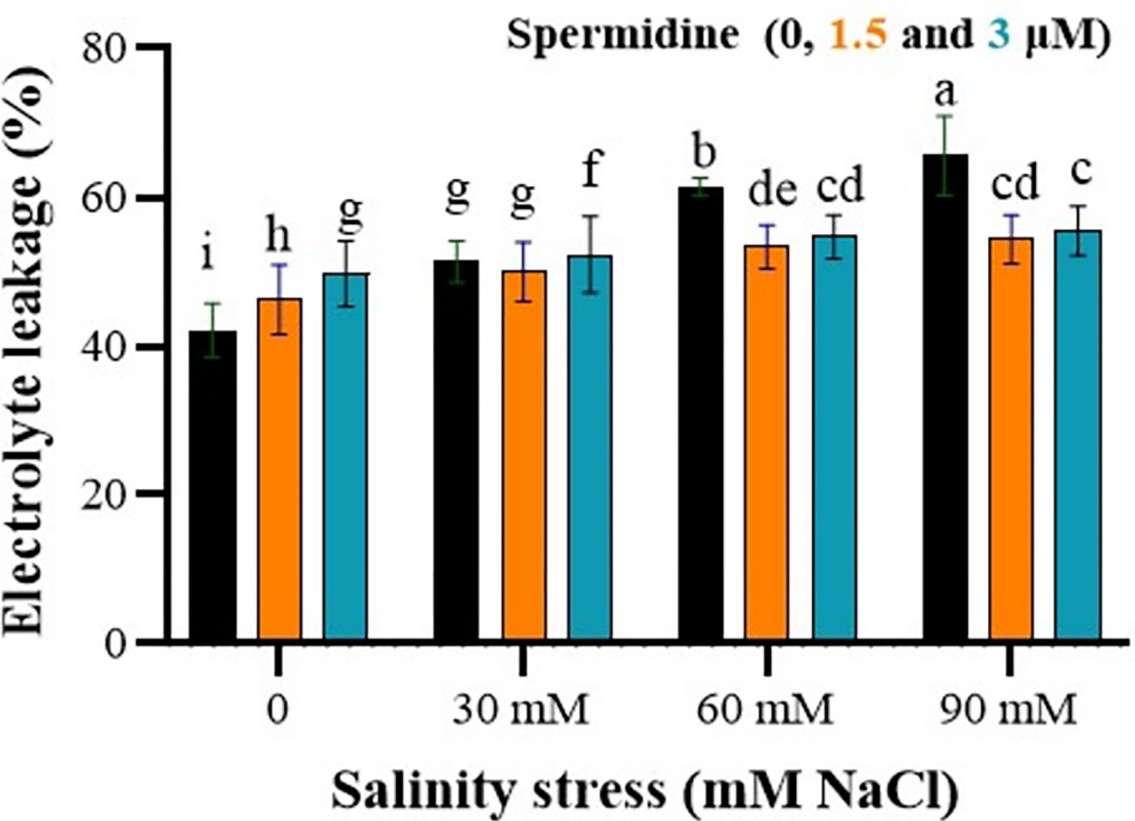

**Fig 2. Interaction effects of spermidine and salinity on electrolyte leakage of yarrow.** Bars represent the standard error, and different letters above bars indicate significance based LSD test (p<0.01).

carotenoids (7.37 mg/g FW) under non-stressed conditions were found in control plants without spermidine application. Compared to control, 1.5 and 3 μM spermidine resulted in lower pigment levels in non-saline soils. Salinity stress dose-dependently decreased chlorophyll a, with the maximum reduction of 68.7% observed at 90 mM NaCl without spermidine. Meanwhile, spermidine application further lowered chlorophyll a at higher salinities, with 3 μM producing the least amount (4.26 mg/g FW) at 90 mM NaCl. In contrast, both spermidine levels increased chlorophyll b and carotenoid contents under most saline conditions relative to untreated stressed plants. Particularly, 1.5 μM spermidine elevated chlorophyll b by 11% and carotenoids by 22% at 90 mM salinity over the non-spermidine control. Collectively, these findings demonstrate spermidine conferred partial protection of the photosynthetic pigment pool against salt-induced degradation, likely to sustain photosynthesis during stress.

## Spermidine enhance proline content under salinity stress

The analysis of variance results revealed a significant impact of salinity stress, spermidine, and their interaction on the proline content in the plants (P < 0.01). As salinity stress and spermidine concentration increased, there was a notable elevation in the proline content (Fig 4). Proline levels in control plants ranged from 133–190 μmol/g FW based on spermidine treatment. Under non-stressed conditions, 1.5 and 3 μM spermidine led to minor proline accumulation compared to control. Salinity stress induced a dose-dependent rise in proline, reaching a maximum value of 0.56 μmol/g FW at 90 mM NaCl with 3 μM spermidine. Both spermidine levels

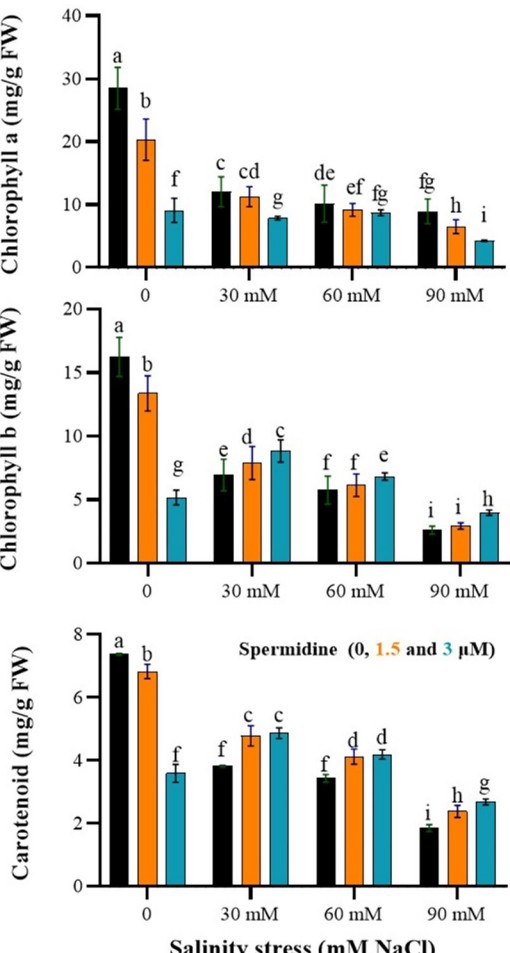

**Fig 3. Interaction effects of spermidine and salinity on photosynthetic pigments of yarrow.** Bars represent the standard error, and different letters above bars indicate significance based LSD test (p<0.01).

further augmented proline content above untreated stressed plants across all salinity levels. At 90 mM salinity, 3 μM spermidine elevated proline by 14.4% over the non-spermidine control. Meanwhile, 1.5 μM spermidine was less efficient than 3 μM in enhancing proline synthesis under saline conditions. These results demonstrate that spermidine application synergistically aided salinity-induced proline accumulation, playing a protective role via its contribution to osmotic adjustment in stressed plants.

### Spermidine improve carbohydrate content under salinity stress

The study found a significant impact of salinity stress, spermidine, and their interaction on the total carbohydrate content. Upon analyzing the mean data, a clear pattern emerged, showing a significant increase in the total carbohydrate content as both salinity stress and spermidine concentration escalated (Fig 5). As shown in Fig 5, control plants had carbohydrate levels ranging from 0.3–0.5 μg/g FW depending on spermidine treatment. Salinity stress dose-dependently elevated carbohydrates from 0.5–0.84 μg/g FW without spermidine. Both spermidine doses further boosted carbohydrate accumulation compared to untreated stressed plants at all salinity levels. At 90 mM NaCl, 3 μM spermidine augmented carbohydrates to the highest

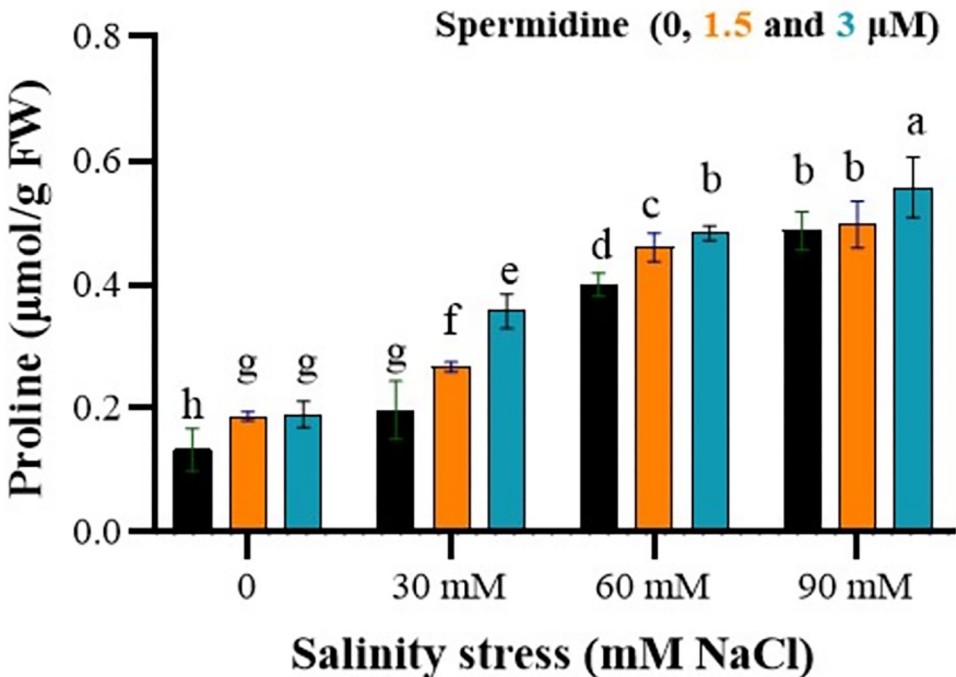

**Fig 4. Interaction effects of spermidine and salinity on proline content of yarrow.** Bars represent the standard error, and different letters above bars indicate significance based LSD test (p<0.01).

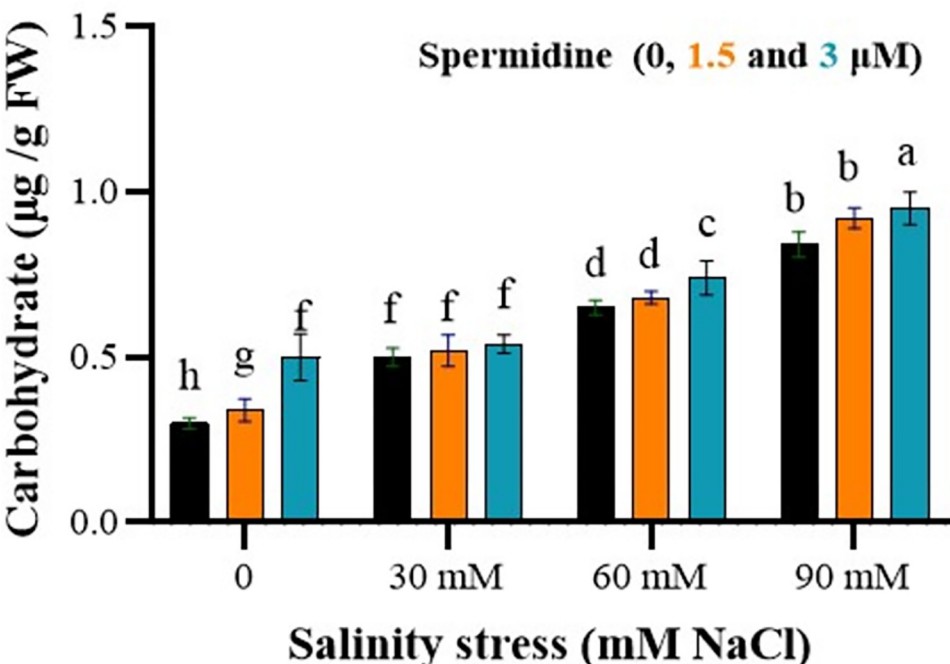

**Fig 5. Interaction effects of spermidine and salinity on carbohydrate of yarrow.** Bars represent the standard error, and different letters above bars indicate significance based LSD test (p<0.01).

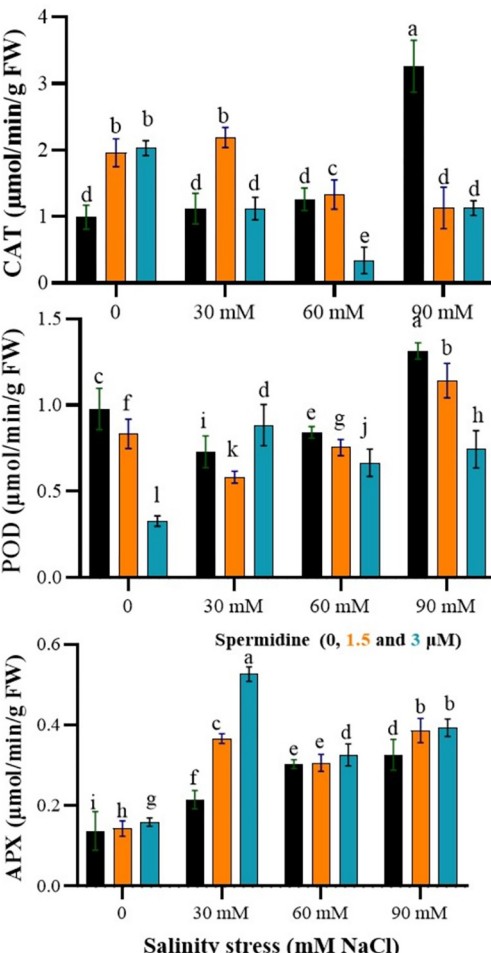

**Fig 6. Interaction effects of spermidine and salinity on antioxidant enzymes (APX, POD, and CAT) of yarrow.**
Bars represent the standard error, and different letters above bars indicate significance based LSD test (p<0.01).

level of 0.95 µg/g FW. Notably, 3 µM spermidine enhanced carbohydrates by 13.1% over the non-spermidine control under 90 mM salinity stress. By comparison, 1.5 µM spermidine's effect was more moderate, accumulating 0.92 µg/g FW carbohydrates at the highest salinity.

## The complex regulation of antioxidant enzymes by spermidine under salinity stress

The results revealed a significant impact of salinity stress, spermidine, and their interaction on APX, POD and CAT activity. As shown in Fig 6, CAT activity in control plants ranged from 0.99–2.03 µmol/min/g FW with varying spermidine levels. Salinity dose-dependently increased CAT, with the maximum of 3.26 µmol/min/g FW at 90 mM NaCl without spermidine. Both spermidine levels augmented CAT above untreated stressed plants except at 60 mM salinity. Notably, 1.5 µM spermidine boosted CAT 1.96-fold in controls over non-spermidine controls. For POD, the highest value of 1.314 µmol/min/g FW was observed at 90 mM salinity without spermidine. Increasing the salinity stress levels to 60 and 90 mM, coupled with the application of spermidine at a concentration of 3 µM, resulted in a reduction in POD enzyme activity in the studied plant species. The lowest POD enzyme activity was recorded whiteout

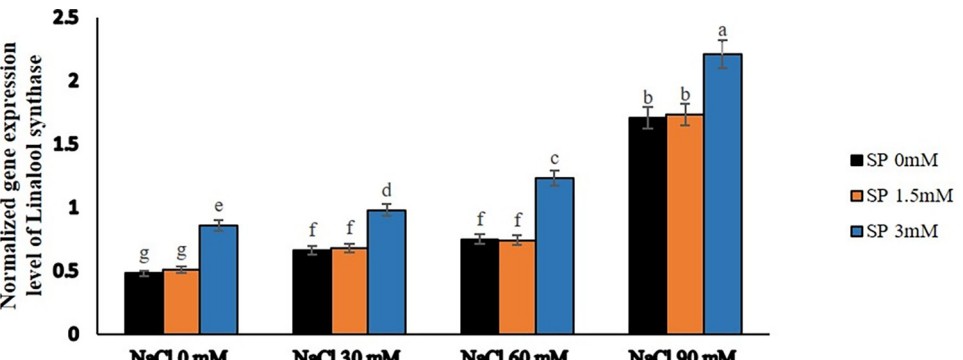

**Fig 7. The effect of spermidine foliar spraying on the expression level of linalool synthase gene in yarrow under salinity stress.** Bars represent the standard error, and different letters above bars indicate significance based LSD test (p<0.01).

salinity applied, accompanied by the application of spermidine at a concentration of 3 μM, yielding an activity of 0.33 μmol/min/g FW. APX activity ranged from 0.137–0.159 μmol/min/g FW in controls. It was observed that the highest activity of the APX enzyme occurred at a salinity level of 30 mM in conjunction with the application of spermidine at a concentration of 3 μM, resulting in an activity of 0.53 μmol/min/g FW. Conversely, the lowest activity was recorded in the control group, with an activity of 0.14 μmol/min/g FW. Salinity and 1.5 μM spermidine generally elevated APX, but 3 μM spermidine performed better, enhancing it 3.85-fold over controls at 30 mM NaCl. Collectively, these results demonstrate spermidine confers antioxidative protection in plants against salinity-imposed oxidative stress, especially at 3 μM.

## Salinity stress and spermidine enhance the expression of linalool and pinene synthetase genes

The results of gene expression showed that using spermidine and exposing plants to salty conditions significantly affected the expression of linalool synthetase and pinene synthetase genes (Figs 7 and 8). Specifically, when spermidine was at a concentration of 1.5 μM, it didn't noticeably impact the expression of these genes. However, spraying a 3 μM spermidine solution on the leaves resulted in a significant increase in the number of gene transcripts.

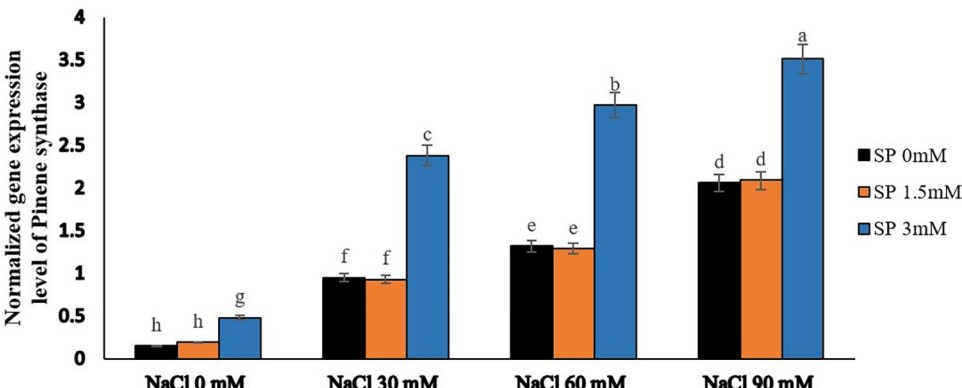

**Fig 8. The effect of spermidine foliar spraying on the expression level of pinene synthase gene in yarrow under salinity stress.** Bars represent the standard error, and different letters above bars indicate significance based LSD test (p<0.01).

At the same time, subjecting the plants to varying levels of salt stress caused an increase in the activity of these genes. This increase was directly related to the intensity of the salt stress. The highest level of gene co-expression occurred at a salt stress level of 90 mM. Further investigation revealed that applying both 90 mM salt stress and spraying a 3 μM spermidine solution on borage plants led to the genes reaching their maximum expression levels. On the contrary, the lowest expression levels were seen in the control group (without salt stress) when either no spermidine or a 1.5 μM spermidine concentration was used.

## Discussion

The results of this study demonstrated that the electrolyte leakage increased with the elevation of salinity stress. At salinity levels of 30, 60, and 90 mM, the application of 1.5 μM spermidine concentration led to a reduction in electrolyte leakage. The stresses induced a gradual increase in ethylene production, subsequently reducing membrane stability and increasing membrane fluidity, which correlated with increased electrolyte leakage [33]. Polyamines, including spermine, are known to inhibit ethylene production. The application of polyamine spermine protects pine seedlings against oxidative stresses and delays the senescence of leaves in drought-stressed seedlings. It is suggested that antioxidant compounds and anti-ethylene agents can help maintain membrane integrity and their physiological functions under stress conditions [33]. Additionally, external application of spermine has been found to improve the condition of rice plants under salinity stress by strengthening the membrane, scavenging free radicals, and maintaining the potassium to sodium ratio [34]. Polyamines, due to their polycationic nature, can interact with negatively charged phospholipids present between lipid bilayers or other macroanions, such as nucleic acids and proteins, thereby contributing to the biological stability of membranes and macro-molecular structures in cells [35].

The study findings revealed a decrease in the content of photosynthetic pigments with increasing salinity stress. The reduction in chlorophyll a and b under stress conditions can be attributed to enhanced pigment degradation, decreased synthesis, and disruption of the enzymes responsible for pigment synthesis. Additionally, the decrease in specific membrane proteins and the increased activity of chlorophyllase and peroxidase enzymes may also contribute to the reduction of chlorophyll under stress conditions [36]. Furthermore, the reduction in leaf greenness may be partly due to reduced nitrogen flow to tissues and alterations in the activity of enzymes like nitrate reductase. Mesophyll conductance is considered the primary limiting factor for photosynthesis in stress-exposed plants [37]. The decline in photosynthesis caused by salinity stress and subsequent drought stress is influenced by both stomatal conductance and mesophyll conductance [38]. Moreover, a decrease in specific membrane proteins under stress conditions and an increase in the activity of chlorophyllase and peroxidase enzymes can also contribute to the reduction of chlorophyll under stress conditions [37]. Carotenoids serve as accessory pigments in chloroplasts, aiding in light absorption. However, their crucial role lies in their ability to mitigate the toxicity of various forms of activated oxygen generated during the excitation of photosynthetic compounds by light. Carotenoids harness energy from excited molecules and free oxygen, converting it into thermal energy. Carotenoids exhibit relatively lower sensitivity to stress compared to chlorophylls, although some plants have shown decreased carotenoid content under water and salinity stress [37]. Application of spermidine at a concentration of 1.5 μM resulted in an increase in photosynthetic pigment content. Polyamines contribute to the reduction of chlorophyll degradation and enhance light capture to improve photosynthetic rates, although the precise molecular mechanism remains unclear. The positive effect of polyamines on chlorophyll content might be attributed to their

antioxidant properties, which prevent the degradation of chloroplast membrane structure [39].

The results of this study showed that under salinity stress, the content of proline increased, with the highest level observed at 90 mM salinity, indicating the plant's resistance to stress conditions. Proline is one of the most important and abundant compatible solutes reported in plants under salinity and drought stresses, and its content variation is a dominant phenomenon in these conditions, which helps in osmotic stress modulation and protects the cell by scavenging ROS [40]. In fact, the increase in proline content under salinity stress may be due to the accumulation of sodium ions in high amounts in the cytosol, which becomes toxic. Salt toxicity in plants under salinity stress is actually a response to the reduction of water potential in the root environment, which leads to leaf water content reduction and intensified salt stress. This results in lipid peroxidation and disruption of the function and structure of cell membranes. Consequently, it stimulates the activity of glutamine kinase, the first enzyme in the proline biosynthesis pathway, leading to an increase in the level of this amino acid [41]. The accumulation of proline in the cytoplasm occurs to maintain water potential balance, and it has the least inhibitory effect on cell growth among all amino acids. By reducing the osmotic potential of root cells, it provides the necessary conditions for water and nutrient uptake [42]. Therefore, considering the effective role of proline amino acid in mitigating the detrimental effects of environmental stresses, especially salinity and osmotic regulation, this increase is justifiable. Spraying spermidine resulted in an increase in proline content, which is consistent with the findings of Ghanbari et al. [43] on squash plants. Polyamines may have multiple effects. For example, in addition to scavenging free radicals, they can affect the activity of antioxidant enzymes and also influence proline. Furthermore, these substances as growth regulators can reduce $Na^+$ and $Cl^-$ during drought stress [44].

The total carbohydrate content showed a significant increase with increasing salinity levels, such that the highest amount was observed at a salinity level of 90 mM and under foliar application of 3 and 1.5 µM spermidine concentrations. One of the main ways plants adapt to saline environments and regulate osmotic balance and water uptake in cells is through the accumulation of organic osmolytes such as carbohydrates. Plant carbohydrates, when exposed to stresses like salinity, can act as protective and osmoregulatory compounds, storing carbon and scavenging free radicals [45]. Accumulated sugars under salinity stress assist in establishing and maintaining cellular structures and functions by interacting with cellular macromolecules, including enzymes [46]. Additionally, this increase may be attributed to the ability of tolerant cultivars to maintain stomatal openings and sustain photosynthesis under salinity stress conditions [47].

The present study investigated the impact of increasing salinity stress on CAT enzyme activity, revealing a significant increase under these conditions. Salinity, along with other non-biological stresses, induces oxidative stress. The elevated activity of the APX enzyme under stress is attributed to the rise in reactive oxygen species, which trigger signaling pathways that lead to the upregulation of antioxidant enzyme genes and subsequent enhancement of enzyme activity [48]. Antioxidants play a crucial role in maintaining cellular oxidative balance by directly interacting with various active oxygen species and scavenging them. When the activity of certain antioxidant enzymes decreases, other enzymes compensate by increasing their activity levels [23]. Researchers have identified the increased levels of antioxidants as a key factor in plant resistance to environmental stresses. The heightened activity of antioxidant enzymes involved in defense mechanisms helps mitigate damage to cell membranes, DNA, and protein oxidation [49]. Consequently, during salinity-induced stress, the levels of peroxidase (POD) compounds rise, followed by an increase in both the activity and concentration of antioxidant

enzymes under drought stress, which aids in the acceleration of defense against free radicals [50].

In this context, spermidine has demonstrated a significant effect on the activity of antioxidant enzymes, including CAT, POD, and APX, under salinity stress conditions, resulting in increased tolerance to drought stress. When sprayed at a concentration of 1.5 μM, spermidine increased CAT activity under 30 mM and 60 mM salinity stress, while a concentration of 3 μM increased POD activity under 30 mM salinity stress. The application of polyamines enhances the activity of antioxidant enzymes, thereby protecting plants against various environmental stresses and reducing their adverse effects. External application of spermidine has been reported to increase CAT activity, thereby protecting cells against free radicals and various stresses [51, 52]. CAT and POD play pivotal roles in neutralizing hydrogen peroxide toxicity by converting it into water and molecular oxygen, thus preventing cellular damage under unfavorable conditions like salinity stress [53]. The antioxidant effect of polyamines primarily stems from their cationic properties at physiological pH or their ability to improve ion balance, enabling them to scavenge free radicals and inhibit lipid peroxidation [54]. Free polyamines bind to macromolecules, safeguarding them against oxidative damage, while their role in cellular osmotic balance and pH regulation is also noteworthy [55]. Certain characteristics of polyamines resemble those of antioxidants in terms of maintaining cell membrane stability and mitigating the effects of hydrogen peroxide. Changes in polyamine biosynthetic genes lead to the production of biosynthetic polyamines [28]. The abundant production of polyamines arises from reciprocal reactions, with polyamines freely interacting with anionic molecules such as DNA, RNA, proteins, and lipid membranes, providing protection under stressful conditions [56]. A previous study reported a significant increase in CAT and superoxide dismutase enzyme activity in cyclamen plants following spermidine foliar application [57].

Gene expression studies demonstrated that the expression of linalool synthetase and pinene synthetase genes was significantly influenced by spermidine and exposure to salty conditions. Notably, the application of a 3 μM spermidine solution on plants resulted in a substantial increase in gene transcripts, whereas a concentration of 1.5 μM spermidine had negligible effects on gene expression. Furthermore, subjecting plants to varying levels of salt stress enhanced the activity of these genes, with the highest co-expression observed at a salt stress level of 90 mM. These findings were supported by other studies indicating the involvement of these genes and their products in plant responses to stress. For instance, Zhang et al. [58] observed differential expression of terpene synthase genes, including linalool synthase, in response to cold stress in S. album. Similarly, Senji and Abdollahi Mandoulakani [59] found an increase in linalool synthase gene expression under cold stress conditions. These observations suggested that these genes contributed to plant defense against abiotic stresses such as cold.

Terpenoids, including linalool and E-β-farnesene, have been implicated in plant defense against both biotic and abiotic stresses. Zhang et al. [58] proposed that these compounds served as signal molecules to attract insects for pollination or possessed antioxidant properties that helped mitigate the negative consequences of stress, such as scavenging reactive oxygen species (ROS). The overproduction of monoterpenes and sesquiterpenes, such as linalool, was suggested as an antioxidant-based defense mechanism against oxidative stress induced by water deficit [22]. Moreover, the expression of the linalool synthase gene was positively correlated with linalool accumulation in basil plants experiencing drought stress [60]. This implied that linalool biosynthesis represented a mechanism that responded to water-deficit stress, potentially serving as an adaptive response dependent on the cultivar. Additionally, the accumulation of linalool was associated with plant defense against bacterial and fungal pathogens. Shimada et al. [61] detected high levels of linalool in the leaves of Ponkan mandarin, a species

resistant to citrus canker, while susceptible species exhibited minimal linalool content. These findings suggested that linalool biosynthesis and accumulation contributed to plant defense mechanisms. Monoterpenes, including pinene, played a role in plant stress responses. These compounds might have acted as signaling molecules or participated in protecting plants against environmental stressors such as cold and drought.

## Conclusion

The present study demonstrated that salinity stress, spermidine, and their interaction had significant effects on various physiological and molecular parameters in the plants. Salinity stress resulted in increased electrolyte leakage and decreased levels of photosynthetic pigments, while the application of spermidine at specific concentrations mitigated these effects to some extent. The proline content and total carbohydrate content were also influenced by salinity stress and spermidine concentration, with higher levels observed under more severe stress conditions. Moreover, the activity of antioxidant enzymes showed a complex pattern, with variations depending on salinity levels and spermidine application. Finally, gene expression analysis revealed that both spermidine and salt stress played a role in modulating the expression of linalool and pinene synthetase genes, with the highest expression observed under the combined treatment of 90 mM salt stress and 3 μM spermidine application. These findings contribute to our understanding of the physiological and molecular responses of plants to salinity stress and highlight the potential role of spermidine in mitigating the adverse effects of such stress.

## Supporting information

**S1 Data.**
(XLSX)

## Author Contributions

**Conceptualization:** Mohammad-Reza Raji.

**Formal analysis:** Mostafa Farajpour.

**Investigation:** Sajedeh Alijani, Mohammad-Reza Raji, Zohreh Emami Bistgani, Abdollah Ehtesham Nia.

**Methodology:** Sajedeh Alijani, Zohreh Emami Bistgani.

**Project administration:** Mohammad-Reza Raji.

**Supervision:** Mohammad-Reza Raji, Zohreh Emami Bistgani, Mostafa Farajpour.

**Writing – original draft:** Sajedeh Alijani, Mostafa Farajpour.

**Writing – review & editing:** Mohammad-Reza Raji, Zohreh Emami Bistgani, Abdollah Ehtesham Nia, Mostafa Farajpour.

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
