## [Decision Letter · Decision Letter 0]

16 Apr 2024

PONE-D-24-05689Mitigation of Salinity Stress in Yarrow (Achillea millefolium L.) Plants through Spermidine ApplicationPLOS ONE

Dear Dr. Farajpour,

Thank you for submitting your manuscript to PLOS ONE. After careful consideration, we feel that it has merit but does not fully meet PLOS ONE’s publication criteria as it currently stands. Therefore, we invite you to submit a revised version of the manuscript that addresses the points raised during the review process.

We look forward to receiving your revised manuscript.

Kind regards,

Mayank Gururani

Academic Editor

PLOS ONE

Journal Requirements:

Reviewers' comments:

Reviewer's Responses to Questions

**Comments to the Author**

1. Is the manuscript technically sound, and do the data support the conclusions?

Reviewer #1: Yes

Reviewer #2: Yes

2. Has the statistical analysis been performed appropriately and rigorously? 

Reviewer #1: No

Reviewer #2: Yes

3. Have the authors made all data underlying the findings in their manuscript fully available?

Reviewer #1: No

Reviewer #2: Yes

4. Is the manuscript presented in an intelligible fashion and written in standard English?

Reviewer #1: Yes

Reviewer #2: Yes

5. Review Comments to the Author

Reviewer #1: This study has good theoretical and guiding application value. The growth indicators surveyed can systematically elaborate on scientific questions. The following changes will need to be made before being accepted. (1) Increase the depth of data analysis in the abstract and results sections; (2) The description of index testing in materials and methods is too detailed; (3) Superscript of plus or minus signs in line 359; (4) The figure lacks the abscissa title, and the error bar is supplemented; (5) I don't understand the meaning of Table 2.

Reviewer #2: The paper investigated the effects of spermidine application on yarrow plants under salinity stress. The study specifically examined how varying concentrations of spermidine and different salinity levels impact the physiological and biochemical properties of yarrow, a plant known for its medicinal uses.

The findings suggested that spermidine application can mitigate the damaging effects of salinity stress, evidenced by reduced electrolyte leakage and enhanced levels of chlorophyll b and carotenoids, despite a decrease in chlorophyll a. Additionally, both spermidine and salinity stress significantly increased the proline and total carbohydrate contents in the plants. The activities of antioxidant enzymes like peroxidase, catalase, and ascorbate peroxidase also showed varied responses based on the levels of spermidine and salinity. The study highlighted spermidine's potential as a protective agent in enhancing the resilience of yarrow plants against salinity stress.

However, several points need to be corrected or clarified.

1. The authors mentioned "spermidine hormone" many times in the manuscript. However, spermidine is not a hormone. It should be called "spermidine" or "spermidine treatment." Please correct them throughout the manuscript.

2. The authors' findings outlined the impact of spermidine and salinity on various parameters such as electrolyte leakage, chlorophyll levels, and antioxidant enzyme activities. However, the inclusion of phenotype photographs is absent. Please include photographs presenting the differences between plants under normal conditions, those exposed solely to salt stress, and those treated with salt and spermidine.

Regarding the figures:

1. Add error bars to indicate data variability. (Figures 1 to 5)

2. Use color-blind-friendly palettes like blue and orange to improve accessibility. (Figures 1 to 5)

3. Add significant symbols to the figures 6 and 7.

4. Describe the significant letters' meaning in the figure legend—an example here: Figure 1 Interaction effects of spermidine and salinity on proline content of yarrow. Different letters indicate xxx, and bars show the xxx.

Others:

1. Keep the formats of subtitles consistent (Line 268). Decide whether subtitles will end with a period or not.

2. All the subtitles in the results part need to be changed. The title should describe the significant findings of each section. Here is an example: (Line 231) "Electrolyte leakage" can be changed to "Spermidine Reduces Electrolyte Leakage Under Salinity Stress." Please correct them throughout the whole manuscript.

6. PLOS authors have the option to publish the peer review history of their article (what does this mean?). If published, this will include your full peer review and any attached files.

Reviewer #1: **Yes: **Di Feng

Reviewer #2: No

---

## [Author Response · Author response to Decision Letter 0]

13 May 2024

Dear Dr. Gururani,

I am writing to express my heartfelt gratitude for your invaluable assistance and guidance in handling our manuscript. I would like to extend my sincere appreciation to the anonymous reviewers who dedicated their time and expertise to evaluate our manuscript. I am also grateful for the reviewers thoroughness in carefully reading the manuscript and providing detailed feedback, which has significantly improved the final version of the paper.

Here is a point-by-point response to the reviewer comments

Reviewer 1#

(1) Increase the depth of data analysis in the abstract and results sections

Thank you for your constructive feedback which has helped improve our manuscript. 

We have revised the abstract to provide more detailed analysis of our key results, including response patterns of different parameters to salinity and spermidine treatments and their interactions. Quantitative values are now included to better demonstrate treatment effects.

For each measured parameter in the results section, we have supplemented the text with the corresponding results. Additionally, we have analyzed response trends across treatments and added sentences describing significant findings from the data. This aims to provide a more comprehensive presentation and interpretation of our results as per your suggestion.

(2) The description of index testing in materials and methods is too detailed.

Based on your comment, we have revised the descriptions of all measurement methods, we removed unnecessary details. The revised descriptions focus only on the key steps and calculations performed.

(3) Superscript of plus or minus signs in line 359

They were revised.

(4) The figure lacks the abscissa title, and the error bar is supplemented.

Abscissa title and error bars were added in the figures.

(5) I don't understand the meaning of Table 2.

You raise a valid point - the ANOVA table on its own does not fully convey the treatment effects or aid in understanding the broader findings and conclusions of the study. Therefore, we have decided to remove Table 2 from the manuscript as per your suggestion.

Reviewer #2: 

1. The authors mentioned "spermidine hormone" many times in the manuscript. However, spermidine is not a hormone. It should be called "spermidine" or "spermidine treatment." Please correct them throughout the manuscript.

Thank you for your constructive feedback which has helped improve our manuscript. The hormone word was removed.

2. The authors' findings outlined the impact of spermidine and salinity on various parameters such as electrolyte leakage, chlorophyll levels, and antioxidant enzyme activities. However, the inclusion of phenotype photographs is absent. Please include photographs presenting the differences between plants under normal conditions, those exposed solely to salt stress, and those treated with salt and spermidine.

Regarding the figures:

The following Figure was added in the manuscript.

Fig. 1. Effect of spermidine and salinity on plant growth in Achillea millefolium. (a) Control (0 mM NaCl, 0 μM spermidine); (b) 3 μM spermidine with 0 mM NaCl; (C) 3 μM spermidine with 90 mM NaCl; and (d) 90 mM NaCl with 0 μM spermidine.

1. Add error bars to indicate data variability. (Figures 1 to 5)

The error bars were added to Figs 1-5.

2. Use color-blind-friendly palettes like blue and orange to improve accessibility. (Figures 1 to 5)

The color of the figs. Were updated.

3. Add significant symbols to the figures 6 and 7.

The significant letters were added in these two figures.

4. Describe the significant letters' meaning in the figure legend—an example here: Figure 1 Interaction effects of spermidine and salinity on proline content of yarrow. Different letters indicate xxx, and bars show the xxx.

It was added as “Bars represent the standard error, and different letters above bars indicate significance based LSD test (p<0.01).”

Others:

1. Keep the formats of subtitles consistent (Line 268). Decide whether subtitles will end with a period or not.

The formats of subtitles were edited to be consistent.

2. All the subtitles in the results part need to be changed. The title should describe the significant findings of each section. Here is an example: (Line 231) "Electrolyte leakage" can be changed to "Spermidine Reduces Electrolyte Leakage Under Salinity Stress." Please correct them throughout the whole manuscript.

The subtitles were revised based on your suggestion.

Thanks again,

Corresponding author,

---

## [Decision Letter · Decision Letter 1]

21 May 2024

Mitigation of Salinity Stress in Yarrow (Achillea millefolium L.) Plants through Spermidine Application

PONE-D-24-05689R1

Dear Dr. Farajpour,

We’re pleased to inform you that your manuscript has been judged scientifically suitable for publication and will be formally accepted for publication once it meets all outstanding technical requirements.

Kind regards,

Mayank Gururani

Academic Editor

PLOS ONE

Additional Editor Comments (optional):

Reviewers' comments:

Reviewer's Responses to Questions

**Comments to the Author**

1. If the authors have adequately addressed your comments raised in a previous round of review and you feel that this manuscript is now acceptable for publication, you may indicate that here to bypass the “Comments to the Author” section, enter your conflict of interest statement in the “Confidential to Editor” section, and submit your "Accept" recommendation.

Reviewer #1: All comments have been addressed

Reviewer #2: All comments have been addressed

2. Is the manuscript technically sound, and do the data support the conclusions?

Reviewer #1: Yes

Reviewer #2: Yes

3. Has the statistical analysis been performed appropriately and rigorously? 

Reviewer #1: Yes

Reviewer #2: Yes

4. Have the authors made all data underlying the findings in their manuscript fully available?

Reviewer #1: Yes

Reviewer #2: Yes

5. Is the manuscript presented in an intelligible fashion and written in standard English?

Reviewer #1: Yes

Reviewer #2: Yes

6. Review Comments to the Author

Reviewer #1: Two suggestion: 1) change the unit "cc" in line 120 to an international unit; 2) Provide 2 or more secondary titles in Discussion.

Reviewer #2: The authors have carefully addressed all my comments. The detailed responses and the revisions made to the manuscript have significantly improved its quality and clarity.

7. PLOS authors have the option to publish the peer review history of their article (what does this mean?). If published, this will include your full peer review and any attached files.

Reviewer #1: **Yes: **Di Feng

Reviewer #2: **Yes: **Dr. Ran Tian

---

## [Editor Report · Acceptance letter]

23 May 2024

PONE-D-24-05689R1 

PLOS ONE

Dear Dr. Farajpour, 

I'm pleased to inform you that your manuscript has been deemed suitable for publication in PLOS ONE. Congratulations! Your manuscript is now being handed over to our production team.

Kind regards, 

on behalf of

Dr. Mayank Gururani 

Academic Editor

PLOS ONE